# The Role of Glycosyltransferases in Colorectal Cancer

**DOI:** 10.3390/ijms22115822

**Published:** 2021-05-30

**Authors:** Cecilia Fernández-Ponce, Noelia Geribaldi-Doldán, Ismael Sánchez-Gomar, Roberto Navarro Quiroz, Linda Atencio Ibarra, Lorena Gomez Escorcia, Ricardo Fernández-Cisnal, Gustavo Aroca Martinez, Francisco García-Cózar, Elkin Navarro Quiroz

**Affiliations:** 1Department of Biomedicine, Biotechnology and Public Health, Faculty of Medicine, University of Cadiz, 11003 Cadiz, Spain; ceciliamatilde.fernandez@uca.es (C.F.-P.); ismael.sanchez@uca.es (I.S.-G.); curro.garcia@uca.es (F.G.-C.); 2Institute of Biomedical Research Cadiz (INIBICA), Cadiz 11009, Spain; noelia.geribaldi@uca.es (N.G.-D.); ricardo.fernandez@inibica.es (R.F.-C.); 3Department of Human Anatomy and Embryology, Faculty of Medicine, University of Cadiz, 11003 Cadiz, Spain; 4Center of Mathematics, Computing and Cognition (CMCC) Laboratory of Computational Biology and Bioinformatics—LBCB, Federal University of ABC, Sao Paulo 09210-580, Brazil; roberto.navarro@ufabc.edu.br; 5Faculty of Medicine, Simón Bolívar University, Barranquilla 080001, Colombia; lindatencioi@gmail.com (L.A.I.); garoca1@hotmail.com (G.A.M.); 6Clínica de la Costa, Barranquilla 080001, Colombia; lgomez@clinicadelacosta.co; 7Centro de Investigación e Innovación en Biomoléculas, Care4you S.A.S., Barranquilla 080001, Colombia; 8Faculty of Basic and Biomedical Sciences, Simón Bolívar University, Barranquilla 080001, Colombia; 9School of Medicine, Fundación Universitaria San Martin, Puerto Colombia 080002, Colombia

**Keywords:** colorectal cancer (CRC), glycosyltransferase, glycosylation, post-translational modification

## Abstract

Colorectal cancer (CRC) is one of the main causes of cancer death in the world. Post-translational modifications (PTMs) have been extensively studied in malignancies due to its relevance in tumor pathogenesis and therapy. This review is focused on the dysregulation of glycosyltransferase expression in CRC and its impact in cell function and in several biological pathways associated with CRC pathogenesis, prognosis and therapeutic approaches. Glycan structures act as interface molecules between cells and their environment and in several cases facilitate molecule function. CRC tissue shows alterations in glycan structures decorating molecules, such as annexin-1, mucins, heat shock protein 90 (Hsp90), β1 integrin, carcinoembryonic antigen (CEA), epidermal growth factor receptor (EGFR), insulin-like growth factor-binding protein 3 (IGFBP3), transforming growth factor beta (TGF-β) receptors, Fas (CD95), PD-L1, decorin, sorbin and SH3 domain-containing protein 1 (SORBS1), CD147 and glycosphingolipids. All of these are described as key molecules in oncogenesis and metastasis. Therefore, glycosylation in CRC can affect cell migration, cell–cell adhesion, actin polymerization, mitosis, cell membrane repair, apoptosis, cell differentiation, stemness regulation, intestinal mucosal barrier integrity, immune system regulation, T cell polarization and gut microbiota composition; all such functions are associated with the prognosis and evolution of the disease. According to these findings, multiple strategies have been evaluated to alter oligosaccharide processing and to modify glycoconjugate structures in order to control CRC progression and prevent metastasis. Additionally, immunotherapy approaches have contemplated the use of neo-antigens, generated by altered glycosylation, as targets for tumor-specific T cells or engineered CAR (Chimeric antigen receptors) T cells.

## 1. Introduction

Colorectal cancer (CRC) has been categorized worldwide as the third most common diagnosed cancer and the fourth most usual cause of cancer death [1]. The etiological heterogeneity, pathogenesis and risk evolution factors of CRC are not well defined [2,3]. Environmental, genetic, epigenetic and post-translational modifications (PTMs), as CRC risk factors, have been the focus of several research efforts, with the aim of improving not only the molecular treatments used in these patients but also biomarker profiles that could be useful for early diagnosis and a better patient follow-up during and after treatment [4].

Among the post-translational modifications differentially identified in CRC tissues, mechanisms of glycosylation have been extensively studied. Alterations of glycan biosynthesis associated with abnormal glycosylation patterns of several molecules have been related with oncogenesis, cell growth, apoptosis inhibition, increased motility, cell migration, cell adhesion, tumor cell invasiveness and metastasis. Glycans act as interface molecules between cells and microenvironment and in several cases glycan structures confer functions to the molecules. Thus, tumorigenesis and metastasis could be narrowly associated with glycan profile changes in cancer cells [5,6,7,8].

In cancer tissues, glycan expression has been found upregulated, downregulated and, interestingly, glycan structures have been shown to be truncated or modified exhibiting new structures [5,6]. Thus, the study of alterations in glycosylation profiles of molecules with a role in cancer progression and metastasis could partially explain molecular mechanisms underlying malignization and even serve as biomarkers.

Glycosylation of molecules, such as proteins or lipids, takes place in the endoplasmic reticulum and Golgi apparatus in a process catalyzed in a non-templated manner by glycosyltransferases, such as fucosyltransferases, sialyltransferases, and N-acetylglucosaminyltransferases in association with glycosidases. Proteins can be N-glycosylated and O-glycosylated and these variations determine, in part, their physical and functional properties. The considerable weight of lipid glycosylation in biology has also been demonstrated [9,10,11]. As the amount and glycosyltransferase profile could be different depending on the cell type, glycoconjugates can also be dissimilar. In fact, several biological functions are influenced by glycan biosynthetic pathways and glycosylation processes determined by the cell conditions [12]. In addition, environmental and genetic conditions have shown an impact in the glycome, thereby generating phenotypic variations [13]. In order to design new treatments aimed at specifically modifying the glycome in pathologic tissues, it is important to extend the understanding of tissue-specific glycosylation [14,15].

Downregulation or upregulation of glycosyltransferase expression, unavailability of glycosylation cofactors or substrates, changes in glycosyltransferase subcellular localization and mutations in genes encoding glycosyltransferases have been evidenced in cancer samples. These features generate aberrant glycosylation signatures that are related with tumorigenesis, metastasis and even with chemoresistance [16,17].

Accordingly, mutations in glycosyltransferases and glycosylation pathways associated genes, as well as altered gene expression coding glycosyltransferases have been described for different CRC cell lines and tumor tissues from several stages [18]. Additionally, differential glycoproteomics and glycolipid profiles have been identified and analyzed. Molecules such as annexin A1, mucins, heat shock protein 90, β1 integrin, selectin ligands, carcinoembryonic antigen (CEA), epidermal growth factor receptor (EGFR), insulin-like growth factor-binding protein 3 (IGFBP3), transforming growth factor beta (TGF-β) receptors, Fas (CD95), PD-L1, sorbin and SH3 domain-containing protein 1 (SORBS1), decorin, CD147 and glycosphingolipids show variations in their glycosylation pattern in CRC tissues when they are compared with healthy tissues. Interestingly, these changes are associated with the prognosis and evolution of the disease and response to radiation or chemotherapy agents [10,19,20]. Thus, studies aimed to identify altered glycosyltransferase gene expression together with glycosylation profile of molecules involved in CRC pathogenesis may provide a molecular basis for the identification of predictive biomarkers of treatment effectiveness and new therapeutic targets.

In this review, we intend to summarize the role of glycosyltransferases and glycosylation profile in CRC pathogenesis and discuss their therapeutic implications.

## 2. Glycosyltransferase Gene Expression Profile in Colorectal Cancer

The expression pattern of glycosyltransferase genes plays an important role in the biology of CRC cells [21].

Among the genes coding for glycosyltransferases, observed in CRC, 16 have been found upregulated (FUT1, FUT2, FUT3, FUT4, FUT5, FUT6, FUT8, B3GNT3, B3GNT8, B4GALNT3, C1GALT1, MGAT4B, MGAT5A, OGT, ST6GAL1 and ST6GALNAC1) and 16 have been found downregulated (FUT9, B3GNT1, B3GNT6, GALNT6, GCNT3 MGAT3, MGAT5B, ST3GAL1, ST3GAL3, ST6GALNAC2, ST6GALNAC3, ST6GALNAC6, ST8SIA1, ST8SIA3, ST8SIA4 and ST8SIA5) (Table 1) [22,23,24,25,26,27,28,29,30,31,32].

The upregulation of genes coding for beta 3 and beta 4 glycosyltransferases (B3GNT3, B3GNT8, C1GALT1 and B4GALNT3), fucosyltransferases (FUT1, FUT2, FUT3, FUT4, FUT5, FUT6 and FUT8), mannosyl-glycoprotein N-acetylglucosaminyltransferases (MGAT4B and MGAT5A), O-linked N-acetylglucosaminyltransferases (OGT) and sialyltransferases (ST6GAL1 and ST6GALNAC1) plays a crucial role in tumor cell proliferation, survival, induction of stem-like cell properties, epithelial–mesenchymal transition (EMT), metastasis and resistance to chemotherapy and radiotherapy (Table 1).

On the other hand, the downregulation of glycosyltransferases expression in CRC cells, such as beta 3-glycosyltransferases (B3GNT1 and B3GNT6), polypeptide N-acetylgalactosaminyltransferase (GALNT6), glucosaminyl (N-acetyl) transferases/xylosyltransferases (GCNT3), fucosyltransferases (FUT9), mannosyl-glycoprotein N-acetylglucosaminyltransferases (MGAT3, MGAT5b) and sialyltransferases (ST3GAL1, ST3GAL3, ST6GALNAC2, ST6GALNAC3, ST6GALNAC6, ST8SIA1, ST8SIA3, ST8SIA4 and ST8SIA5) has been also associated with tumor progression, tumor growth, cell proliferation, adhesion, migration, invasion and chemoresistance to 5-Fluorouracil (5-FU) (Table 1).

Regarding beta 3-glycosyltransferases, the role of β3GnT8 (UDP-GlcNAc:betaGalbeta-1,3-N-acetylglucosaminyltransferase 8) in CRC cell metastasis could be explained by decorating CD147 with β1,6-branched polylactosamine structures [22,27,33]. CD147, a cell surface glycoprotein also known as extracellular matrix metalloproteinase inducer (EMMPRIN) in its highly glycosylated active form, promotes matrix metalloproteinase (MMP) production in stromal and tumor cells, generates extracellular matrix degradation and thereby increases tumor cell migration and invasion [33,34]. Additionally, it has been evidenced that the overexpression of CD147 in CRC cells promotes cancer stem cell (CSC) maintenance, EMT, metastasis and low sensitivity to 5-FU through MAPK/ERK signaling pathway activation [35,36]. Furthermore, other beta-3 glycosyltransferase upregulated in CRC cells known as C1GALT1 (core 1 β1,3-galactosyltransferase) promotes cell invasion, stem-like cell phenotype and radioresistance via modifying the O-glycosylation pattern of FGFR2 (fibroblast growth factor receptor 2) and β1 integrin, respectively (Table 1) [26,37].

The upregulation of mannosyl-glycoprotein N-acetylglucosaminyltransferases, observed in CRC cells has implications in tumor progression and metastasis (Table 1). The overexpression of N-acetylglucosaminyltransferase V (MGAT5A), evidenced in CRC, promotes cancer cell migration and invasiveness through aberrant glycosylation of TIMP-1 (tissue inhibitor of metalloproteinase-1) [38]. Additionally, MGAT5A activity is involved in the preservation of angiogenesis in anti-VEGF refractory tumors through the induction of galectin 1 (Gal 1) binding to endothelial cells [39].

Alterations in the addition of fucose to precursor glycan structures promote cell proliferation, migration, adhesion to extracellular matrix, invasiveness, metastasis, maintenance of CSCs, TGF-β-induced EMT and multidrug-resistance (MDR) [40,41,42,43,44,45]. Oligosaccharides are fucosylated by FUTs in order to synthesize Lewis antigens, which are involved in cell adhesion to endothelial cells, tumor metastasis, tissue differentiation and inflammation. The overexpression of FUT3 and FUT5, evidenced in cancer cells, correlates with the increased presence of Lewis antigens on cancer cell surface that facilitates carcinoma cell–endothelial cell interactions, tumor cell rolling and metastasis [44]. Cancer cell migration and invasion are also promoted by TGF-β-induced EMT and are stimulated by fucosylation of TGF-β receptors. The overexpression of FUT3, FUT6 and FUT8 correlates with the enhanced TGF-β downstream signaling and consequently with increased cell invasiveness and metastasis [42,45]. In addition, alterations in cell surface fucosylated oligosaccharides are associated with cancer cell multidrug resistance (MDR) not only mediated by EMT but also stimulated by the positive regulation of drug efflux through ABC transporters via aberrant activation of PI3K/Akt signaling pathway. Interestingly, increased expression of FUT4, FUT6 and FUT8 has been associated with upregulation of PI3K/Akt signaling pathway and high levels of MRP1 (multidrug resistance-related protein 1) [41]. By contrast, FUT9 activity shows a dual role in CRC progression. At later stages of CRC, downregulation of FUT9 favors cancer cell proliferation and migration. On the other hand, at earlier stages of CRC development, FUT9 expression promotes CRC cell reprogramming towards a stem cell-like phenotype contributing to the expansion of CSCs or tumor-initiating cells (TICs) [46,47].

Sialylation is also a process related to CRC. Modifications in the expression of sialyltransferases are associated with cancer cell survival, proliferation, migration, invasion, metastasis, induction of stem-like cell properties, chemotherapy resistance, increased sialyl-Tn antigen expression, inflammation-driven carcinogenesis and resistance to 5-FU. In this respect, overexpression of ST6GAL1 and ST6GALNAC1 has a role in cancer cell stemness, cell survival, metastasis and chemoresistance [22,48,49,50,51]. ST6GALNAC1 induces stem-like cell properties via activation of AKT pathway, while ST6GAL1 has been evidenced to induce the expression of stem cell transcription factors Sox9 and Slug, thereby promoting cancer cell stemness. In addition, stem-like cell phenotype confers metastatic properties as well as resistance to drugs [48,50]. Furthermore, sialylation of Fas, catalyzed by ST6GAL1, inhibits Fas receptor internalization and, thus, avoids Fas-mediated apoptosis [51]. On the other hand, downregulation of ST6GALNAC2, observed in CRC tissue, stimulates metastatic processes (Table 1). ST6GALNAC2 hampers the interaction of soluble galectin-3 with the O-glycan profile on the cell surface through the modification of O-glycans. Thus, ST6GALNAC2 suppresses the retention of tumor cells at secondary sites and avoids metastasis. Consequently, downregulation of ST6GalNAc2 generates unmodified O-glycans expressed at the cell surfaces that interacts with Galectin-3 and facilitates the development of metastasis [52]. In addition, the downregulation of ST8SIA4, evidenced in CRC and other cancer cells, has been associated with increased cell proliferation, migration and invasion (Table 1) [22,53]. It has been described that ST8SIA4 gene expression is downregulated by mir-146a and miR-146b, which are overexpressed in thyroid carcinoma [53]. Similarly, epigenetic mechanisms such as DNA methylation and histone modification regulate glycosyltransferase gene expression [54]. In cancer cells, alterations of miRNAs expression profile and epigenetic mechanisms are involved in the regulation of glycosyltransferase gene expression, and glycosylation in turn participates in the epigenetic modulation of histones and transcription factors; thus, glycosylation could be considered an integral part of the epigenetic code [54].

Accordingly, changes in glycosyltransferase gene expression evidenced in carcinogenesis show consequences in glycosyltransferase joint activity and, therefore, in the cell glyco-code. Some of these alterations affect several molecular functions related with tumorigenesis, metastasis and cancer progression.

**Table 1 ijms-22-05822-t001:** Upregulated and downregulated glycosyltransferase genes in CRC.

**A. Upregulated glycosyltransferase genes in CRC**. Cellular effects induced by the upregulation of genes encoding glycosyltransferases are described.			
**Glycosyltransferases**	**Gene**	**Effects on Cancer Cells**	**References**
Beta 3-glycosyltransferases	B3GNT3	Cell migration. Cell invasion. Maintenance of CSCs.	Ashkani 2016 [22]. Barkeer 2018 [55].
	B3GNT8	Cell migration. Cell invasion. Resistance to 5-FU.	Ashkani 2016 [22]. Ishida 2005 [27]. Ni 2014 [33]. Shen 2014 [56].
	C1GALT1	Induction of stem-like cell properties. Cell survival. Cell migration. Cell invasion. Radioresistance.	Hung 2014 [26]. Zhang 2018 [37].
Beta 4-glycosyltransferases	B4GALNT3	Cell migration. Cell invasion. Maintenance of CSCs.	Che 2014 [23].
Fucosyltransferases	FUT1	Cell proliferation. Cell migration. Cell invasion. Metastasis. EMT. Maintenance of CSCs.	Ashkani 2016 [22]. Lai 2019 [43]. Petretti 2000 [57].
	FUT2	Cell proliferation. Cell adhesion to extracellular matrix. Cell migration. Cell invasion. Metastasis. EMT. Maintenance of CSCs.	Ashkani 2016 [22]. Lai 2019 [43].
	FUT3	TGF-β-induced EMT. Cell adhesion to endothelium. Cell migration.	Ashkani 2016 [22]. Meng 2017 [58]. Padró 2011 [44]. Hirakawa 2014 [42].
	FUT4	MDR.	Ashkani 2016 [22]. Cheng 2013 [41]. Petretti 2000 [57].
	FUT5	Cell adhesion to endothelium. Cell migration.	Ashkani 2016 [22]. Padró 2011 [44].
	FUT6	TGF-β-induced EMT. MDR. Tumor progression. Metastasis.	Ashkani 2016 [22]. Cheng 2013 [41]. Hirakawa 2014 [42]. Sethi 2014 [31].
	FUT8	Tumor progression. Cell migration. Cell invasion. Metastasis. TGF-β-induced EMT. Tumor immune evasion. EGF-mediated cellular growth.	Sethi 2014 [31]. Tu 2017 [45]. Bastian 2021 [40].
Mannosyl-glycoprotein N-acetylglucosaminyl transferases	MGAT4B	Tumor progression. Metastasis.	Ashkani 2016 [22].
	MGAT5A	Tumor progression. Metastasis. Cell invasion. ↓ Anti-VEGF effectivity. Increase CCSC population.	Murata 2000 [29]. Guo 2014 [59]. Kim 2008 [38]. Croci 2014 [39]. Petretti 2000 [57].
O-linked N-acetylglucosaminyl transferases	OGT	Cell proliferation. Cell migration. Cell invasion.	Xu 2019 [32].
Sialyltransferases	ST6GAL1	Cell migration Cell invasion. Cell survival. Induction of stem-like cell properties. Chemotherapy resistance.	Ashkani 2016 [22]. Schultz 2016 [50]. Swindall 2011 [51]. Park 2012 [49]. Sethi 2014 [31].
	ST6GALNAC1	↑Sialyl-Tn expression. Maintenance of CSCs. Resistance to 5-FU.	Ashkani 2016 [22]. Marcos 2004 [60]. Ogawa 2017 [48].
**B. Downregulated glycosyltransferase genes in CRC.** Cellular effects induced by the downregulation of genes encoding glycosyltransferases are described.			
**Glycosyltransferases**	**Gene**	**Effects on Cancer cells**	**References**
Beta 3-glycosyltransferases	B3GNT1		Ashkani 2016 [22].
	B3GNT6	Cell migration. Cell invasion. Metastasis. EMT.	Iwai 2005 [28]. Gupta 2020 [61].
Polypeptide N-acetylgalactosaminyl transferases	GALNT6	Poor differentiation. Cell migration. Cell invasion. Chemoresistance to 5-FU. ↑Tn-antigen expression.	Noda 2018 [30].
Glucosaminyl (N-acetyl)transferases/ xylosyltransferases	GCNT3	Cell proliferation. Cell adhesion. Cell migration. Cell invasion. Cell survival. Tumor growth. Chemoresistance to 5-FU.	Huang 2006 [25]. González-Vallinas 2015 [24]. Fernández 2018 [16].
Fucosyltransferases	FUT9	Cell migration. Metastasis.	Ashkani 2016 [22]. Auslander 2017 [46].
Mannosyl-glycoprotein N-acetylglucosaminyl transferases	MGAT3	Tumor progression. Metastasis.	Ashkani 2016 [22].
	MGAT5b	Tumor progression. Metastasis.	Ashkani 2016 [22].
Sialyltransferases	ST3GAL1		Ashkani 2016 [22].
	ST3GAL3	Tumor progression. Metastasis.	Ashkani 2016 [22]. Sethi 2014 [31].
	ST6GALNAC2	Metastasis.	Ashkani 2016 [22]. Murugaesu 2014 [52]. Ferrer 2014 [62].
	ST6GALNAC3	Tumor progression.	Ashkani 2016 [22]. Haldrup 2018 [63].
	ST6GALNAC6	Inflammation-driven carcinogenesis.	Ashkani 2016 [22]. Huang 2020 [64].
	ST8SIA1		Ashkani 2016 [22].
	ST8SIA3		Ashkani 2016 [22].
	ST8SIA4	Cell proliferation. Cell migration. Cell invasion.	Ashkani 2016 [22]. Ma 2017 [53].
	ST8SIA5		Ashkani 2016 [22].

TGF-β: Transforming growth factor β. EMT: Epithelial–Mesenchymal Transition. MDR: Tumor Multidrug Resistance. CCSC: Colon Cancer Stem Cells. CSCs: Cancer Stem Cells. EGF: Epidermal Growth Factor. 5-FU: 5-fluorouracil. Tn-antigen: Cancer associated truncated glycan.

## 3. Glycosylated Molecules in Colorectal Cancer

Glycosylation is associated with several biological processes. Alterations in glycosyltransferase levels and glycosylation patterns have been evidenced in inflammatory conditions, tumorigenesis and metastasis [5,6,7,8,65,66]. In tumor cells, the increased dysregulation in glycosylation patterns induced, in part, by changes in cancer cell glucose metabolism favors their growth and metastasis [7,67,68].

Glycosylation of proteins generates changes on their biophysical properties, function, distribution and retention in the plasma membrane and modulates cell behavior, cellular interactions, specific ligand–receptor interactions and immune recognition [5,69,70,71]. In CRC tissues, highly glycosylated membrane proteins, such as annexin A1, have been identified (Table 2). Annexin A1 shows increased levels of GlcNAcylation in CRC compared to healthy tissues [20]. Annexin A1 is a calcium-dependent phospholipid-binding protein that interacts mainly with intracellular phospholipidic components such as phosphatidylserine and has been associated with several biological processes involved in oncogenesis, such as vesicle aggregation, membrane aggregation, exocytosis, endocytosis, actin polymerization, mitosis, cell membrane repair, apoptosis and cell differentiation. In addition, other annexin A1 functions include protection against myocardial ischemia-reperfusion injury; and the regulation of inflammatory processes, such as leukocyte migration and activation, as well as the activity of phospholipase A2 (PLA2), cyclooxygenase-2 (COX-2) and inducible nitric oxide synthase (iNOS) [72,73]. The increased GlcNAcylation described in annexin A1 from CRC tissues is defined as the addition of N-acetylglucosamine (GlcNAc) moieties to serine/threonine residues of the protein catalyzed by O-GlcNAc transferases (OGT) [74]. The importance of O-GlcNAcylation in several cellular events and even in tumorigenesis could be explained, in part, by its role in gene expression. In this regard, O-GlcNAcylation has been associated with transcription factor stability and functions. Thus, O-GlcNAcylation of transcription factors is necessary for T and B lymphocyte activation. Likewise, OGT cooperates with the regulation of epigenetic mechanisms, such as histone modification and DNA methylation that have been found dysregulated in cancer cells [74,75]. Furthermore, O-GlcNAc modifications have roles in glucose metabolism, cell differentiation, cell–cell adhesion, mitosis, Ca^2+^ signaling and the regulation of other ion channels among other cellular processes [76]. Thus, glycosylation of annexin A1 could have a role in CRC etiology and progression in addition to being a consequence of the carcinogenesis process.

Another protein found differentially GlcNAcylated in CRC tissues is heat shock protein 90 (HSP90β) (Table 2) [20]. HSP90β is a chaperone protein that interacts with enzymes, transcription factors, co-chaperones, structural proteins and oncoproteins among others molecules implicated in a plethora of cellular events [77]. In tumorigenesis, HSP90β has been associated with cell viability [78]; HSP90β post-translational modifications (PTMs), such as phosphorylation, acetylation and S-nitrosylation have been implicated in cancer cell immortalization via modulation of telomerase activity [79]. Additionally, it has been shown that HSP90 O-GlcNAcylation affects proteasome activity and, likewise, HSP90 glycation alters HSP90 activity and could be related with cancer progression [80].

Glycosylation of mucins has been extensively studied in cancer tissues. In this regard, MUC1 is highly glycosylated in CRC tissues and the O-glycosylation pattern of MUC2 is altered (Table 2) [19,81]. Mucin expression in intestinal epithelium is modulated by different factors, such as gut microbiota, dietary components, epigenetic mechanisms and translational processes. Since mucins are one of the most important components of the gastrointestinal associated immune system, alterations in mucin expression or in their structure and functional properties due to changes in N-glycosylation and O-glycosylation patterns among other PTMs have shown a role in immune dysregulation, loss of mucosal barrier integrity, increased risk of inflammatory or infectious diseases and changes in cell adhesive properties that alter the interaction between cells and their microenvironment [82]. In this regard, the CRC risk increases in patients affected by inflammatory bowel diseases (IBD), which are characterized by a marked dysbiosis, anomalous immunological response against gut microbiota, epigenetic dysregulation and alterations in the expression of enzymes associated with PTMs [83]. Thus, mucins’ contribution to IBD and CRC pathogenesis has been studied by several research efforts. MUC1 is one of the mucins that is more strongly implicated. Physiologically, decreased MUC1 expression in colon mucosa stimulates CD4+ T cell polarization to the Th17 phenotype and in turn Th17 cells produce cytokines that stimulate MUC1 expression, thus limiting the inflammation condition. According to these findings, alterations in MUC1 structure, such as glycosylation, can abrogate its immunoregulatory functions and ensues in chronic inflammatory conditions [84]. In addition, MUC1 contributes to metastasis generation by facilitating cell–cell and cell–extracellular matrix interactions due to its capacity to bind several molecules associated with migration and extravasation found the presence of both at the cell surface and in the cytoplasmic compartment. Thus, MUC1 can stimulate metastatic progression not only by promoting cell–cell adhesion but also by activating intracellular signaling pathways [85]. Interestingly, mucin oligosaccharides are a source of nourishment for microbiota and participates in the maintenance of colonic flora balance and distribution. Therefore, alterations in mucin glycans can generate dysbiosis and consequently abrogate gut immune balance and intestinal mucosal integrity [83]. In addition, a dense and diverse O-glycosylation pattern of mucins, such as MUC2 described as the main gel-forming mucin, is a key protective factor against colitis and CRC through the supply of health nourishment for the gut microbiota, maintenance of mucus bacterial adhesion capacity and protection from digestive proteases [19,86]. Thereby, mucin glycosylation patterns are critical not only for mucin structure but also for its stability, functionality and protection. Therefore, aberrant mucin glycosylation has strong implications in IBD and CRC pathogenesis and progression.

Adhesion molecules, such as selectin, integrin and their respective ligands, have an essential role in tumor cell endothelial adhesion, migration and organ invasion [87]. Changes in the expression level and glycosylation pattern of adhesion molecules have been related with tumor progression, poor prognosis and metastasis. In this regard, the hypersialylation of β1 integrin evidenced in CRC cells (Table 1) facilitates β1 integrin binding to its ligands, collagen I and laminin and interestingly promotes intracytoplasmic association of β1 integrin with talin, stimulating cell adhesion, motility and migration [88]. Studies focused on selectins have been performed with respect to the ability of circulating tumor cells to spread to distant organs and lymph nodes. In CRC, the overexpression of E-selectin ligands, specifically the sialyl LewisX (sLeX) determinants that mediate cell rolling, have been described. Thus, metastasis is closely associated with an altered glycosylation profile [89].

Regarding the immune response, alterations of the glycosylation profile of mucins or other molecules affect tumor cells interactions with human lectin receptors on antigen presenting cells (APCs), such as MGL (macrophage galactose-type lectin), DC-SIGN (dendritic cell-specific intercellular adhesion molecule-3-grabbing non-integrin) and galectin-3. In CRC tissues, aberrant glycosylation of MUC1 and other CRC antigens, such as carcinoembryonic antigen (CEA), can alter the mammalian lectin receptor interactions with tumor cells and could participate in immune modulation and immune evasion (Table 2) [81]. In addition, glycosylation pattern of T cells and their ligands, determine thymus homing of T cell precursors, T cell migration, endothelial adhesion and T cell interaction with MHCII on APC. Additionally, N-glycosylation of T cell receptor (TCR) and T cell co-receptor regulates T cell activation through stabilization of immunological synapses and inhibition of proteases degradation. Moreover, CD4+ T cell polarization is modulated, in part, by the expression and function of glycosyltransferases [90].

Additionally, in CRC, CEA shows high levels of Lewis antigens such as Lewis X and Lewis Y. These carbohydrates mediate the binding of CEA with DC-SIGN which is expressed mainly on immature dendritic cells. The interaction of tumor cells with immature dendritic cells through DC-SIGN could be related with immune tolerance due to the fact that immature dendritic cells are not effective in priming naive T cells. Thus, recruitment of immature dendritic cells by DC-SIGN through Lewis antigens decorating CEA can arrest dendritic cell differentiation towards a mature phenotype, thereby contributing to tumor cell immune evasion (Table 2) [81,91].

Aberrant glycosylation of the epidermal growth factor receptor (EGFR) has been widely studied in CRC cells (Table 2). N-linked glycosylation of EGFR is required for its trafficking to the plasma membrane and it is necessary for its role in tumor growth [92]. In addition, the EGFR glycosylation pattern in CRC cells is modified by β1,4-N-acetylgalactosaminyltransferase III (B4GALNT3) that has been found overexpressed in advanced CRC stages and has been related with poor prognosis (Table 1). PTMs catalyzed by B4GALNT3 confer stability to EGFR and avoids its degradation, contributes with EGFR signaling pathways that mediate the maintenance of stemness and consequently generates cancer cell growth, survival and attenuated differentiation (Table 2) [23].

Furthermore, insulin-like growth factor-binding protein 3 (IGFBP3) has shown an altered glycosylation pattern in serum and tumor cell membrane samples isolated from CRC patients. In CRC, IGFBP3 has been found decorated with high amounts of α2,6 Sialic acid and low levels of Fuc and GlcNAc moieties [93]. The main function of IGFBP3 is to protect and transport the insulin-like growth factors (IGFs), in circulation, thereby regulating IGFs activity and availability. IGFs play a role in cell proliferation, survival, differentiation and migration. In addition, IGFBP3 has shown biological associations with cell growth inhibitors and molecules involved in tumorigenesis such as retinoic acid, retinoid X receptor (RXR), retinoic acid receptor (RAR), nuclear factor kappa B (NF-ĸB), transforming growth factor beta (TGF-β), tumor necrosis factor-alpha (TNFα), glucose-regulated protein 78 (GRP78), butyrate (histone deacetylase inhibitor), dietary supplements with anti-inflammatory and anti-cancer properties, caveolin-1 and interestingly with the tumor suppressor gene p53 [94,95,96,97,98]. In addition, the glycosylation pattern of IGFBP3 is crucial for IGFBP3 affinity, its selective binding with its ligands and for IGFBP3 stability, half-life, function and activity. Thus, low expression of IGFBP3 has been related with cancer development and resistance to radiotherapy and chemotherapy and interestingly alterations in the glycosylation pattern of IGFBP3 can modify several molecular events and intracellular pathways related with tumorigenesis [93,94,95,99].

In CRC tissues, altered glycosylation patterns of glycosphingolipids have been described (Table 1) [10]. Glycosphingolipids are plasma membrane molecules that consist of glycan structures linked to sphingosine and closely related lipids. They are implicated in several events related with membrane integrity and cellular interactions. In cancer cells, they are involved in cell proliferation, cell migration, chemoresistance, EMT, metastasis, reduced apoptosis and poor cell differentiation [9,10]. Interestingly, CRC tissues have shown an increased fucosylation and sialylation accompanied by low acetylation and sulfation of glycosphingolipid glycans (Table 2) [10,100]. Thus, PTMs affecting glycosphingolipid glycan profile in tumor tissues could be implicated in CRC progression and metastasis.

**Table 2 ijms-22-05822-t002:** Summary of molecules identified to be glycosylated in colorectal cancer samples. Glycosylation and associated biological effects on cancer cells have been collected.

Target Molecules	Glycosylation	Effects on Cancer Cells	References
Annexin A1	GlcNAcylation	Mitosis. Apoptosis. Cell differentiation.	Li et al. [20]. Yang et al. [74].
HSP90	GlcNAcylation	Cell viability. Cancer progression.	Li et al. [20]. Zou et al. [78]. Overath et al. [80].
Carcinoembryonic antigen (CEA)	↑ Fucose ↑ Mannose ↑ Thomsen–Friedenreich antigen ↓ N-acetylgalactosamine ↓ N-acetylglucosamine ↓ Galactose	Immune tolerance. CRC tumorigenesis. CRC progression.	Zhao et al. [101]. van Gisbergen et al. [91].
IGFBP3	↑Sialylation (α2,3) ↓Fucosylation ↓GlcNAcylation	Cell proliferation. Cell survival. Cell differentiation. Cell migration.	Zámorová et al. [93].
Decorin	O-glycosylation.	CRC development and progression. Metastasis. Cell-cell adhesion. Cell migration.	Wei et al. [102].
SORBS1	O-glycosylation.	CRC development and progression. Metastasis. Cell-cell adhesion.	Wei et al. [102].
EGFR	Sialylation (Loss of α2,6 sialylation) Modification of N- with LacdiNAc structures.	Cell proliferation. Tumor growth. Cancer cell survival. Attenuated cancer cell differentiation. Chemoresistance.	Li et al. [92]. Che et al. [23]. Park et al. [49].
TGF-β receptors	Fucosylation	EMT. Metastasis.	Hirakawa et al. [42].
MucinsMUC1 MUC2	O-glycosylation	Alteration of the interactions of mammalian lectin receptors with tumor cells. Immune dysregulation. Loss of mucosal barrier integrity. Gut dysbiosis.	Pothuraju et al. [82]. Peixoto et al. [69]. Brockhausen et al. [103]. Venkitachalam et al. [17]. Kawashima et al. [19]. Arike et al. [86].
Podoplanin	O-glycosylation	Cell migration. Cell invasion.	Liu et al. [104].
β1 integrin	α2-6 sialylation	Cell adhesion. Cell motility. Cell migration. Tumor cell survival.	Seales et al. [88]. Zhuo et al. [105].
Fas (CD95)	α2-6 sialylation	Anti-apoptotic effect.	Swindall et al. [51].
PD-L1	N-glycosylation	Immune response evasion.	Ruan et al. [106].
CD147	Modification with Beta1,6-branched polylactosamine structures.	Cell migration. Cell invasion. Metastasis.	Ni et al. [33].
Glycosphingolipids	↑ Fucosylation ↑ Sialylation ↓ Glycans acetylation ↓ Glycans sulfation	Cell proliferation. Cell migration. Chemoresistance. EMT. Metastasis. Reduced apoptosis. Poor cell differentiation.	Holst et al. [10]. Misonou et al. [100]. Cumin et al. [9]. Gb3: Distler et al. [107]. Gb4: Park et al. [108]. GCS: Haynes et al. [109]. NEU3: Yamaguchi et al. [110]. GD1a and GM1: Kwak et al. [111]. α-GalCer: Yoshioka et al. [112]. GM3: Chung et al. [113].

EMT: Epithelial–Mesenchymal Transition.

## 4. Implications of Glycosylation and Glycosyltransferases in Colorectal Cancer Therapy

As previously mentioned, CRC is considered one of the most lethal cancer with a high global prevalence [114]. Current treatments include surgical interventions in order to remove primary tumors and to avoid metastasis either before or after chemotherapy or radiotherapy [115,116,117,118]. Despite these treatments, metastasis is a common occurrence in most cases because of the late diagnoses and the problems in surgical elimination [2,119,120,121]. For these reasons, chemotherapy is a good option to reduce death related to CRC.

Chemotherapy agents are varied and there are regimens including only one agent or a combination of effective drugs that affect one or several targets [121]. Indeed, the most frequently used treatment in present times is a combination of 5-fluorouracil (5-FU)/leucovorin (LV, folinic acid) with oxalipatin or irinitecan (camptothecin-11, CPT-11). Other agents include monoclonal antibodies such as cetuximab or bevacizumab targeting EGFR or the vascular endothelial growth factor (VEGF), respectively [122].

Modulation of glycosyltransferase expression by drugs has been studied for CRC. The expression of GCNT3 gene, which encodes the enzyme mucin-type core 2 1,6-N-acetylglucosaminyltransferase (C2GnT-M), has been found downregulated in bad prognosis CRC tissues (Table 1). In this regard, Gonzalez-Vallinas et al. have shown that the expression of GCNT3 is induced by chemotherapeutic drugs, such as pyrimidine analogue 5-FU, proteasome inhibitor bortezomib or the mitotic inhibitor paclitaxel and these effects correlate with suppression of cell viability in cancer cells [24].

Regarding tumor cell epitopes, it has been described that the altered glycosylation generates neo-antigens that become targets for tumor-specific T cells [90]. Thereby, CRC associated glycosylation can be involved in tumor immune evasion and immune regulation. Based on these findings, immunotherapy challenges, such as those that include the design and generation of engineered CAR (Chimeric antigen receptors) T cells using differentially glycosylated molecules as targets, have ensued (Figure 1) [123]. In addition, innovative immunotherapies focused on enhancing T cell activity through modification of T cell N-glycosylation could avoid the differentiation of naive CD4+ T cells into regulatory or exhaustion phenotypes that favor tumor immune evasion and it is usually observed in precancerous conditions and in tumor microenvironment (Figure 1) [124,125,126,127,128].

According to CRC, O-GlcNAcylation has been shown to be elevated in CRC due to the Warburg effect and it has been recognized as an axis for the development and progression of metastasis [129]. CRC cells replace their aerobic glucose metabolism, even in the presence of relatively normal levels of oxygen, by anaerobic glucose metabolism which leads to a significant increase in glucose absorption in a process known as the Warburg effect [130]. Thus, the Warburg effect provide cancer cells with an incredible adaptive advantage compared to normal cells since glucose and glutamine are basic for cellular growth [131,132]. This process is also associated with a high risk of metastasis in CRC patients who have an excessive caloric intake [133] or diabetes comorbidity [134]. In this regard, various miRNAs have been shown to regulate the expression of O-GlcNAcylation transferases (OGT) [135,136]. Among these, miR-424 downregulates the expression of OGT and other glycosyltransferases, such as MGAT4A, with a resultant arrest of the cell cycle (Figure 1) [137]. In addition, OGT and histone methyltransferase (EZH2) are regulated by miR-101. The evidenced decrease in miR-101 in CRC tissues induces an increase in O-GlcNAcylation of Ser75 that stabilizes EZH2. OGT and EZH2 activity produces a decrease of miR-101 expression, thereby generating a negative feedback loop that promotes metastasis (Figure 1) [138]. Thus, OGT and O-GlcNAcylation patterns are interesting therapeutic targets for metastatic CRC [139].

As previously stated, alterations in glycosylation patterns, mediated by the modified glycosyltransferase expression, are involved in CRC malignant transformation [11,140]. Therefore, glycoconjugates and the molecules involved in their synthesis are interesting targets for new therapeutic approaches. In this context, Jian-Jun Qu et al. evaluated the effect of miR-4262 in CRC cell lines and human tissues and observed a significant decrease of miR-4262 levels when compared with control samples. Additionally, upregulation of miR-4262 expression decreased GALNT4 expression. GALNT4 catalyzes the O-glycosylation of threonine residues 44 and 57 in P-selectin glycoprotein ligand-1 (PSGL-1). Thr57 glycosylation is necessary for PSGL-1/P-selectin interaction, which stimulates tumor growth, tumor cell propagation into the bloodstream and metastasis. Accordingly, the administration of miR-4262 could reduce tumor growth and metastasis through the downregulation of GALNT4 expression (Figure 1) [141,142,143,144].

It has been shown that BGJ398, an OGT inhibitor, can significantly inhibit the activity of core 1 O-Glycan T-Synthase (C1GALT1) that catalyzes the transfer of Gal from UDP-Gal to GalNAc-alpha-1-Ser/Thr of cell surface proteins, such as the fibroblast growth factor receptor 2 (FGFR2), resulting in a significant decrease in the invasive capacity of CRC cells (Figure 1). These data suggest that FGF/FGFR2 signaling pathway is part of the phenotypic changes modulated by C1GALT1 and could be a therapy for CRC (Figure 1) [26].

Regarding tumor immune evasion, it has been described that KYA1797K, a β-catenin inhibitor, stimulates the immune response through the regulation of glycosyltransferase expression. The β-catenin and dolichyl-diphosphooligosaccharide-protein glycosyltransferase subunit STT3 (STT3) are overexpressed in CRC. The expression of STT3 is upregulated by β-catenin. Furthermore, upregulation of both proteins correlates with CRC’s worsened prognosis. In this regard, KYA1797K mediates suppression of β-catenin expression with the subsequent downregulation of STT3. STT3 downregulation decreases the stability and the immunosuppressive activity of programmed death receptor 1 (PD-L1) through the inhibition of PD-L1 glycosylation and consequently increases the population of CD8+ T cells and the expression of granzyme B. Therefore, the inhibition of PD-L1 glycosylation reduces the immune evasion through blocking Wnt/β-catenin/STT3 signaling pathway in CRC cells (Figure 1) [106].

The use of glycans as target molecules for immunotherapy has been extensively studied in cancer cells. Glycan-based cancer vaccines are used to decorate tumors with glycan structures, such as Lewis antigens, in order to enhance the presentation of tumor antigens by APCs, including dendritic cells internalization and antigen presentation, thereby improving tumor-specific T cell activation (Figure 1) [90]. Additionally, therapeutic approaches using engineered CAR-T cells with specificity for glycan epitopes differentially expressed in cancer cells have also been described (Figure 1) [123].

Drug resistance has been associated with aberrant glycosylation profiles. Hamaguchi et al. used glycosylation inhibitors in order to study the relationship between N-glycosylation and drug resistance. Specifically, they observed an effect of swainsonine, an α-mannosidase 2 inhibitor which blocks Golgi oligosaccharide processing, generating altered N-glycan structures that affect 5-FU mechanisms of resistance in CRC cell lines [145]. In the same way, other authors have shown that swainsonine administration improves tumor cell sensitivity towards the apoptotic effect of cisplatin, which results in avoiding drug resistance. These data suggest that inhibition of N-glycosylation could facilitate the cytotoxic action of chemotherapy agents [146].

All these considerations highlight aberrant glycosylation as a crucial issue in CRC treatment and point out some potential targets to improve and design specific therapies.

## 5. Conclusions

Glycosylation is a regulatory mechanism implicated in a number of physiological and pathological processes. The mechanisms of glycosylation are related with several biological events such as membrane integrity, cellular interactions and cell proliferation. In cancer cells, glycosylation has been related to cell migration, chemoresistance, epithelial-to-mesenchymal transition, metastasis, reduced apoptosis, tumor growth and poor cell differentiation. In CRC, alterations in glycoprotein profiles, such as increased sialylation or branched glycan structures and overexpression of fucosylation, have been extensively found. Thereby, characterization and analysis of glycoconjugates have great potential to improve early diagnosis and therapy designs based on new therapeutic targets.

## Figures and Tables

**Figure 1 ijms-22-05822-f001:**
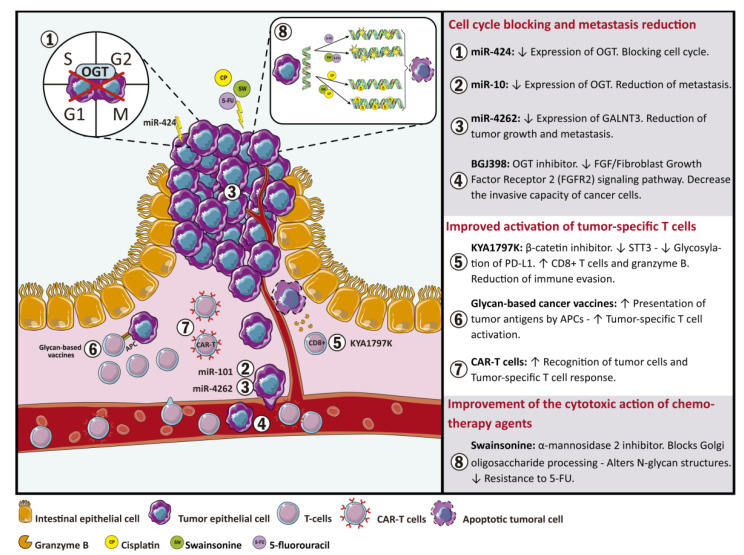
Glycan-based therapeutic approaches for colorectal cancer. Details of therapy strategies based on modifications of glycan and glycosyltransferase expression includes: (1–4) Glycosyltransferases as a target for therapeutic miRNAs and glycosyltransferase inhibitors in order to reduce metastasis and block cell cycle. (5–7) Strategies based on glycan profile modifications to stimulate the immune system. (8) Implementation of modified glycan structures to increase cytotoxic action of chemotherapy agents.

## Data Availability

Data sharing is not applicable to this article.

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
