# Peer review of "The Role of Glycosyltransferases in Colorectal Cancer"

_ijms, 2021, doi:10.3390/ijms22115822_

Round 1
Reviewer 1 Report
In the manuscript titled “The Role of Glycosyltransferases in Colorectal Cancer”, the authors discussed glycosyltransferases and their role in colorectal cancer (CRC). The authors are recommended to make the changes below.
- Since the review is about the role of glycosyltransferases in colorectal cancers, include a separate table for all the glycosyltransferases altered in colorectal cancer and their subsequent oncogenic or tumor suppressor effects. In the main manuscript, authors are recommended to describe in detail the major altered glycosyltransferases in CRC.
- Is EGFR the only main signaling pathway affected by glycosylation in colorectal cancer? There is no description of IGFBP-3 although it is mentioned in table 1.
- The authors are recommended to check for grammatical errors for the flow of the manuscript.
Author Response
We greatly appreciate your suggestions and the careful reading of our manuscript. We have followed every advice in order to improve the redaction and quality of the manuscript.
According to the valuable and helpful recommendations from Reviewer 1, we made the following modifications to the manuscript:
- Separated table for glycosyltransferases upregulated and downregulated in Colorectal cancer and their subsequent oncogenic effects was included as Table 1 (Table has not been highlighted). The altered glycosyltransferase gene expression described in Table 1, was reviewed in detail in “Section 2”: Glycosyltransferase gene expression profile in Colorectal Cancer” of the main manuscript. Modifications in main manuscript have been highlighted.
- Alterations in IGFBP-3 and TGFβ signaling pathway were added to the main manuscript in “Section 3” and “Section 1”, respectively (Modifications have been highlighted).
- English language and grammatical errors were checked and edited (Modifications have been highlighted).
Reviewer 2 Report
Dear Authors,
I would like to suggest some modifications to your article entitled: “The Role of Glycosyltransferases in Colorectal Cancer.” Before being published in the International Journal of Molecular Sciences.
General comments:
- English proofreading
It is imperative that an English native speaker edits your manuscript. Although your review is understandable there are instances of very strange syntax and vocabulary. I will mention just a few:
…Just we mentioned before, CRC… (Just as we mentioned?)
…in most of cases… (most cases or most of the cases)
In cancer samples have been evidenced… (It has been evidenced in cancer samples?)
In CRC tissues have been identified highly glycosylated membrane proteins… (Highly glycosylated proteins have been identified in CRC tissues?)
And so on, apart from strange word choices, overuse of “the”, no number concordance, etc.
- Figures and tables
Figure 1 is not very much informative. Cancer hallmarks are not included. Moreover, the enzymes could have been organized under those processes to which their dysregulation contributes to. If the figure does not become more informative, the upregulation / downregulation of enzymes could be summarized in a table. Maybe add a gene/enzyme table? (see below).
- Nomenclature
There are paragraphs with CC (colon cancer, I guess) instead of the most used CRC.
Consistency with the enzyme/gene names. I read Gnt-V and then in another place name MGAT5, a homogeneous nomenclature should be adopted (maybe the gene name, plus the HUGO for the protein?). It might be useful to consign the enzyme/genes names in tabular form.
- Content organization
Section 2, it is mostly focused on those glycosyltransferases modifying proteins that have been discovered to have a differential glycosylation in CRC (the clearest part of the paper). Then the review shifts to other phenomena involving glycosylation and the focus becomes less and less clear.
I have difficulties with the organization of the information and its placing in different sections. For instance, in section 3, all the GTs found to have an altered expression in CRC are listed. But then only some genes are further talked about. Then different processes/phenomena are mentioned: epigenetic regulation, then glycosylation patterns, then CRC progression and drug resistance, overexpression… When the GT involved in a dysregulated glycosylation is not known, the information is distributed in other sections such as 5 For example: aberrant glycosylations such as mucin type O-glycosylation changes are consigned in section 5 but mucins where discussed before in section 2. Then, Section 3 mentions GCNT3 and drug resistance, why not put it with cancer therapy in section 4?
In summary, I miss a logical thread guiding me through this collection of paper abstracts. So, I suggest a reorganization of the information especially for sections 1, 3, 4 and 5 in order to achieve a more streamlined reading experience.
Specific comments:
I will not suggest any specific comments to this version.
Author Response
We greatly appreciate your suggestions and the careful reading of our manuscript. We have followed every advice in order to improve the redaction and quality of the manuscript.
According to the valuable and helpful recommendations from Reviewer 2, we made the following modifications to the manuscript:
- The manuscript was edited by English native speaker (Modifications have been highlighted).
- According to tables and figure:
- Separated table for glycosyltransferases upregulated and downregulated in Colorectal cancer and their subsequent oncogenic effects was added as Table 1 (Table has not been highlighted). The altered glycosyltransferase gene expression described in Table 1, was reviewed in detail in “Section 2”: Glycosyltransferase gene expression profile in Colorectal Cancer” of the main manuscript. Modifications in main manuscript have been highlighted.
- Table 2. “Summary of molecules identified to be glycosylated in Colorectal Cancer samples” was improved (Modifications have been highlighted).
- Figure 1 was modified following the interesting suggestion of reviewer 2. Figure 1 has been focused in Glycan-based therapeutic approaches for Colorectal Cancer, in order to show in detail the information reviewed in Section 4. Implications of glycosylation and glycosyltransferases in Colorectal Cancer therapy. (Figure 1 has not been highlighted, but modifications associated in main text have been highlighted).
- The nomenclature was checked and edited (Modifications have been highlighted).
- In order to improve the organization of the information, section 5 was eliminated and its paragraphs were including in sections 2, 3 and 4. The alterations in glycosyltransferase gene expression was reviewed in detail, and summarize in a table, in order to obtain a more precise and extensive description (Section 2 – Table 1).
Some paragraphs from section 1 (Introduction) were relocated in sections 2, 3 and 4. Every section was reviewed in detail, in order to ensure a logical sequence.
Round 2
Reviewer 1 Report
In the updated version of the review manuscript titled " The Role of Glycosyltransferases in Colorectal Cancer” the authors have taken into accounts the reviewers' comments and incorporated the recommended corrections. The review would be interesting to readers in cancer biology.
Author Response
We are very grateful for the comments and recommendations. All of them have substantially enriched and improved our manuscript.
We have made the modifications suggested for the reviewer. We would like to say that the reviewer has made a detailed and valuable review of the manuscript. Following his recommendations, we improved the redaction of the manuscript, we checked and modified the capitalized names of molecules and the format of the paragraphs including blank lines and indentations. Modifications are marked up using the “Track Changes” function.
Please find enclosed the revised manuscript. We submit the manuscript entitled “The Role of Glycosyltransferases in Colorectal Cancer.”
Thank you in advance for your consideration of our manuscript. We are open for any further consideration.
Reviewer 2 Report
You'll find the Comments for Authors attached in a Word file.
Author Response

(The authors gave the same response as above.)
